# Employing automatic content recognition for teaching methodology analysis in classroom videos

**Muhammad Aasim Rafique**[1], **Faheem Khaskheli**[2], **Malik Tahir Hassan**[3],
**Sheraz Naseer**[2], **Moongu Jeon**[1]*

1 School of Electrical Engineering and Computer Science, Gwangju Institute of Science and Technology (GIST), Gwangju, Republic of Korea, 2 Department of Computer Sciences, School of Systems and Technology, University of Management and Technology, Lahore, Pakistan, 3 Department of Software Engineering, School of Systems and Technology, University of Management and Technology, Lahore, Pakistan

* mgjeon@gist.ac.kr

**Data Availability Statement:** All relevant data are within the manuscript.

**Funding:** This work was supported by Institute of Information & communications Technology

## Abstract

A teacher plays a pivotal role in grooming a society and paves way for its social and economic developments. Teaching is a dynamic role and demands continuous adaptation. A teacher adopts teaching techniques suitable for a certain discipline and a situation. A thorough, detailed, and impartial observation of a teacher is a desideratum for adaptation of an effective teaching methodology and it is a laborious exercise. An automatic strategy for analyzing a teacher's teaching methodology in a classroom environment is suggested in this work. The proposed strategy recognizes a teacher's actions in videos while he is delivering lectures. In this study, 3D CNN and Conv2DLSTM with time-distributed layers are used for experimentation. A range of actions are recognized for a complete classroom session during experimentation and the reported results are considered effective for analysis of a teacher's teaching technique.

## 1 Introduction

Quality education is one of the seventeen sustainable development goals of the department of economic and social affairs of the United Nations. A pivotal countenance of quality education is a skilled and erudite teacher. Quality teaching is essential for quality education and a teacher herself improves on skills like communication, delivery, enthusiasm, confidence, gestures, and so on. Hence, an authentic and thorough evaluation of teaching skills in a classroom environment and auxiliary feedback is essential. Teaching is a dynamic role that demands adaptation in teaching styles as courses and audiences change. A complete assessment while sitting in a classroom lecture is an exhausting exercise. However, recording of video lectures is a norm and religiously practiced during the recent pandemic time. These recorded videos can be used conveniently for the analysis of the teaching methods of a teacher. A contemporary and

Planning & Evaluation (IITP) grant funded by the Korea government(MSIT) (No.2014-3-00077, AI National Strategy Project).

**Competing interests:** The authors have declared that no competing interests exist.

effective strategy is to employ advancements in artificial intelligence for automated teaching style analysis in recorded video lectures.

A resolve for automated analysis of teaching style in recorded videos is automatic content recognition (ACR), and a retrospective approach is action recognition in the recorded videos. Action recognition in videos is a challenging problem and many of its challenges arise due to the change in viewpoint, camera motion, the scale of objects, pose of the person, change in lighting condition, and background change. Automatic action recognition in videos is a long-standing problem in computer vision. However, thriving solutions provide desirable results in specific domains such as surveillance [1], entertainment, content-based video retrieval [2], Human-Computer Interaction and Robotics [3]. However, a general solution may not work as humans and their actions are not universal. Thus, automatic analysis of teaching style in videos is a specialized ACR problem with its peculiarities.

Recorded video is a sequence of images with implicit branched into a spatial image and temporal sequence of images. The two data are often evaluated with different techniques suitable for exploiting the spatial features in images and capturing the relation in a sequence of images in video datasets. Convolutional Neural Networks (CNNs) [4] are state-of-the-art in object detection and classification in images data [5, 6]. CNN is composed of multiple layers of artificial neural networks (ANN) in which each layer is composed of a filter that hierarchically detects features from the images at different scales. Hierarchical composition of layers of filters learn simple features like edges and curves in early layers and higher layers learn abstraction of meaningful components like faces, shapes, etc., present in the training data. Temporal relationship in a sequence of images has been captured in learning parameters using recurrent neural networks (RNN) [7] and extended in recent years with attention modeling [8, 9]. CNN and RNN need a training dataset of videos with labeled sequences. Some of the publicly available datasets are as follows: UCF101 [10], HMDB51 [11], ViHasi [12], MuHavi [13] and BOSS [14].

This study proposes a framework for automated analysis of teaching methods in wake of the pandemic hit the academia along with all fields of life. The main contributions are as follows: a) An ACR framework equipped with two new deep learning architectures with a self-imposing hard inductive bias for the action recognition of a teacher in a recorded video lecture. b) The study proposes statistical fact generation of the teaching methodology connecting education studies and technology advancement. The collected statistics provide feedback that may be beneficial to the educationists for quality teaching. c) This study chooses potent parameters to understand and measure the amount of time spent by a teacher in content delivery, whiteboard usage and student engagements.

This paper is organized as follows: Section 2 contains a survey of related works. We explain our proposed method in section 3, while section 4 discusses experiments and results. In the end, section 5 concludes this study.

## 2 Related work

Sustaining the quality of education requires continuous and efficient analysis of academic processes, learning objectives, and outcomes, as well as the teaching methodologies applied. It is well realized that teachers and their teaching methods influence students learning significantly in a classroom environment. Monitoring of a teacher's teaching methodology can be online (i.e., in a classroom environment), or offline (e.g., through recorded lecture videos). The objective of teachers' monitoring though should be improved students' learning, not criticizing the respectable teachers. Researchers have been using various techniques for human monitoring, action recognition, and behavior analysis.

Prieto et al. [15] contributed to teacher behavior analysis by applying learning algorithms on sensor data. They used sensors such as accelerometers, EEG, and eye-trackers to collect data and use these to generate statistics using the random forest algorithm. Their model showed good accuracy on training data but achieved only 63% accuracy on the test dataset. They further extended their work and classified the cases to a teacher explaining a concept and monitoring work of the students; used gradient boosting tree, and achieved an improved accuracy of 89% on the test dataset. For action recognition, many different sensors have been used to collect data. Biying et al. [16] have surveyed different sensors and their hardware and software limitations. They talk about different sensors used for activity recognition. The sensors used are divided into different categories such as acoustic, electric, mechanical, optical, and electromagnetic.

Videos and image data are available in abundance these days as closed-circuit television (CCTV) cameras and storage has become cheaper. Frequent use of images and video data for human activity analysis in various applications has been noticed. It also inspires action recognition techniques in computer vision (CV), machine learning (ML), and artificial intelligence (AI). Overall, these techniques include two types of approaches: the first type of approach focuses on feature engineering by extracting useful features such as optic flow, sift, surf, manifold learning, and improved dense trajectory (IDT); and the second type of approach uses deep learning techniques with raw videos. Deep learning architectures for object detection popularly include variants of CNN architectures like ResNet [17], EfficientNet [18] and Inception [19]. In contrast to simple image data, video data have temporal (or sequential) aspects as well. The popular deep learning architectures for predictions in such types of sequential data are recurrent networks with gated units [20] and attention-based model [8].

For better readability, the related works have been organized into three subsections. The works that use sensor-based data come first, then the approaches that use feature engineering, then the approaches that use raw input for the analysis.

## 2.1 Sensor based data

Anna et al. [21] used a smartphone embedded with an accelerometer to collect data and detect human activities. They collected a personalized dataset and created a personalized model for each person based on his age, weight, and height. Their model was evaluated on three different accelerometer-based public datasets which are UniMiB-SHAR, MobiAct, and Motion Sense. Two different classifiers were used which are Support Vector Machine and K-Nearest Neighbor. On Subject dependent testing they were able to achieve 84.79% accuracy on UniMiD-SHAR, 45.57% accuracy on MobiAct, and 43.55% accuracy on the Motion Sense dataset. The average accuracy was 57.79%. When doing subject independent tests they were able to achieve a higher accuracy of 70.19%.

Yu Liang et al. [22] used wearable sensors: a triaxial accelerometer and a triaxial gyroscope on 23 different people. These people wore the sensors on ankles and wrists to perform ten different activities of everyday life, and eleven different sports activities in a laboratory environment. The data from the sensors were used in a Wearable Inertial Sensor Network to recognize the activity performed by the user. Their model was able to achieve 98.23% validation accuracy on everyday activities and 99.55% validation accuracy on sports activities.

Abel et al. [23] also used sensors to compute analytics related to educational design methodologies and lecture delivery by teachers. Four different aspects were observed including integration of learning and teaching analytics, analysis of real-time data collected through sensors and devices in the classroom, teacher's digital literacy in analytics, and planning and evaluation

of teaching activities. The major problems they faced were the analysis of results after lecture delivery, and not having standard measures of performance.

Gregor et al. [24] used sensor data to recognize elderly people activity recognition. The dataset for this project was collected from several sensors including 51 motion sensors, 4 item sensors on selected items, 15 door sensors, and 5 temperature sensors installed in different rooms of the apartment. They tried to use all these sensors' data to recognize 8 different activities, i.e., a bed to toilet transition, cleaning, cooking, grooming, shower, sleep, wake up and work. They used a hidden Markov model (HMM) based system to recognize the activities. Finally, they used a dataset from the CASAS project to test their model. The accuracy was 94.52% on individual activities 70.95% over the combination of all concurrent activities.

Jun Huai et al. [25] have used sensor data to recognize basic activities (BA) and transitional activities (TA). They used the public dataset SBHAR to evaluate their method. Fragments between adjacent basic activities have been used to recognize if the activity is a disturbance activity or a transitional activity. They first split the sensor data into segments, then extract the features of activities based on different sized window segmentation. They used the random forest model to classify the activities either as basic activity or transitional activity. Their model was able to achieve 90% accuracy on different duration windows of sensor data.

Besides sensor data, some researchers have also used notes, comments, or audio data collected during a lecture to analyze a teacher's behavior and methodology of lecture delivery. Anmol et al. [26] created a software that uses audio and visual input from a presentation to summarize the entire presentation. Audio of speaker, slides of the presentation, and handwritten notes combined are used to analyze, summarize and log the summary to a server accessible to authorized users. For this project author used a microphone for audio input and a camera for visual input. Google Speech Recognition was used for the audio input to extract text from the audio. RCNN model was used on visual inputs for text detection. Libraries from OpenCV were used for text extraction. Hence, natural language processing concepts were used for the analysis and generating the summarized data.

Zhao et al. [27] analyzed the behavior of a teacher in the online teaching environment. They used comments provided by 1168 students about 9 different teachers from different mainstream live platforms. These comments were processed through Nvivo software. Their results were subjective. They found that teachers need to focus on characteristics including professional attitude, scientific knowledge, logical reasoning, rhythmic language; and teaching behavior such as precise teaching, flexible interaction, and after-school counseling.

Some researchers have focused on students' behavior so that interaction between teacher and student can be improved. Ku Yu-Te et al. [28] introduced a system called ClassFu which focuses on student behavior in-class activities. ClassFU uses image sensors to analyze a student's behavior. By monitoring the classroom's environment and the level of a student's interaction, ClassFu can collect data during online classes as well as during physical classroom setup. Summary of the data is presented to the teacher, which can be used to improve the student and teacher interactions in the future.

RoboThespian is a life-size humanoid robot that was used to teach children of grade 5 to 7 (Igor et al. [29]). This study focuses on human and robot interaction along with how well can a robot teach to a human student. Two different groups of school students were taught in two different classroom environments. After that, effectiveness in learning new science concepts and creating positive perceptions of the robot teacher was evaluated. The results show that the students were able to understand the concept taught by the robot.

## 2.2 Feature engineering approach

Popular feature extraction algorithms include optic flow, sift, surf, manifold learning, and IDT features. Tran et al. [30] used 3D convolution on video data to create a linear model which took multiple frames as input. The model proposed by [30] was trained on the Sport-1M dataset [31] which yielded 82 to 85% accuracy. Tran et al. further proposed the use of handcrafted IDT features which increased the accuracy of their model up to 90%.

Donahue et al. [32] extracted spatial data from videos using convolution layer blocks and used Recurrent Neural Network (RNN) layer based on Long Short Term Memory (LSTM) cells for temporal features. LSTM, proposed by Hochreiter et al. [7], is a more powerful generalization of vanilla RNN with certain differences in cell architecture. Donahue et al. [32] trained their model using raw RGB frame data and optic flow data. On UCF101 dataset, they were able to achieve 82% accuracy.

Sheng et al. [33] used the OpenPose to extract the coordinates of human joints which are key points from an image or video of a person. These key points were used to classify the actions of a teacher using DenseNet. Along with the teacher's pose, facial emotions were also analyzed using the emotion analysis model of Microsoft. However, they used a self-made dataset to test the model.

Chao and Qiushi et al. [34] have used image data of faces to recognize facial expression. They created their dataset of 10 different people to test their models. They used two different approaches to extract features, in the first approach they used the LBP operator to extract facial contours. After which they created a pseudo-3-D model, which was used to create six facial expression sub-regions. They used two different classification algorithms, SVM and Softmax to classify two different types of expressions which are called the basic emotion model and the circumplex emotion model. Their final results prove that eyes and mouth are the major factors when identifying facial expressions.

## 2.3 Raw input approach

Feichtenhofer et al. [35] proposed a two-stream network consisting of spatial and temporal components. They fused two different 3D CNN models, one for the spatial component and one for the temporal component. UCF101 [10] and HMDB51 [11] datasets were used to train and evaluate the model. The best results they were able to achieve were 92% accuracy on UCF101, and 65% accuracy on HMDB51. In another work, Feichtenhofer et al. [36] used spatial and temporal components of video inputs to recognize actions performed in a video. The input was extracted based on the difference between two successive frames of a video. The authors used these differences between multiple frames in a video as a feed to the CNN model constructed using different fusion techniques. They employed different deep neural network (DNN) architectures including their previous two-stream network combined with LSTM, two-stream combined with CNN, two-stream combined with pre-trained ImageNet weights of VGG16 and ST-ResNet (Spatio-Temporal Residual Networks) [37]. However, from all experiments, the ST-ResNet model with Imagenet weights was able to achieve the best results. UCF101 and HMDB51 datasets were used to evaluate the model and the best results reported by Feichtenhofer et al. [36] were 93% accuracy on UCF101 dataset and 66% accuracy on HMDB51 dataset.

Wang et al. [38] created a different model architecture using CNN layers which accepted both spatial and temporal data as input. The proposed model by [38] was also trained and tested on UCF101 and HMDB51 datasets. The authors reported test accuracies of 94% and 68% respectively on USF101 and HMDB51 datasets that are so far the best on the two aforementioned datasets. Ullah et al. [39] also proposed a deep neural network comprising of CNN

and LSTM to do action recognition. The proposed model by [39] used the CNN layer to extract features from each frame. These features were fed to the LSTM layer which was used to understand the connection between each frame in a video. The proposed model was evaluated on UCF101, HMDB51, and YouTube datasets. The model was able to achieve 87% accuracy on HMDB51 dataset, 91% accuracy on UCF101 dataset, and 92% accuracy on the YouTube dataset.

Takuhiro et al. [40] used a video dataset to recognize group activities performed by a human. Their approach is also based on spatial-temporal features. They have also used CNN for the extraction of spatial features and LSTM for the extraction of temporal features. After extracting these features, they have used a fully connected conditional random field (FCCRF) to classify actions performed by people in the video. They used two different datasets; Collective Activity Data-set and Collective Activity Extended Dataset, to evaluate their model. They were able to improve their baseline accuracy over these datasets. Sharma et. al. [41] present a human action recognition dataset in classroom environments, called EduNet. The dataset is a collection of classroom activities from 1 to 12 standard schools. It has 20 action classes containing both teacher and student activities. Authors report the accuracy of a standard I3D-Res-Net-50 model on the EduNet dataset to be 72.3. Sun et. al [42] have also worked on classroom videos and have contributed a dataset but their focus is on students' classroom behavior, not teachers, e.g. a student is listening, yawning, sleeping, etc. Nida et. al. [43] propose a deep learning method for a teacher's activity recognition in a classroom. They develop a dataset IAVID-I (Instructor Activity Video Dataset-I) having nine action classes and report an accuracy of 81.43% of their model on this dataset. Gang et. al. [44] propose a method to recognize eight kinds of teacher behavior (action classes) in an actual teaching scene achieving an accuracy of 81% on their TAD-08 dataset. In comparison to the above-related works, the focus of our work is on a teacher's classroom activities primarily, not the students. These are university classroom videos. We have 11 action classes and the proposed 3DCNN model achieves a high accuracy of 94%.

## 3 Methodology

Our study progresses in five steps, starting with data collection and ending in experimentation and results analysis. Each of these steps is discussed in detail in the coming subsections.

### 3.1 Dataset introduction and preprocessing

Data abundance plays a vital role [45, 46] in state-of-the-art Deep Learning techniques. Although the data is abundantly produced in today's gadget-loaded environments, yet the usable data is scarce. AI techniques need labeled data for supervised learning algorithms. In this study, data are collected from the lecture recordings of CCTV videos of multiple class sessions held at a university campus. The research has been approved by the Research Ethics and Support Committee, University of Management and Technology, Lahore with a signed approval letter RE-016-2021. The consent of the participants visible in CCTV videos was obtained and approved by the committee. The dataset contains ten videos of five different teachers and each video is more than one hour long. The videos are recorded at a frame rate of 25 frames per second (fps), and the size of each frame is 704 pixels in width and 576 pixels in height (704 x 576). The lecture videos are split into three seconds clips, i.e., 75 frames in each data chunk; and there are around 1050 clips (after removing noisy data). The images from the dataset are augmented using horizontal flip, zoom-in, zoom-out, change of brightness, image rotation and image blur. images from the data Clips generated from the videos are manually annotated with labels of a teacher's actions from the following categories:

- Standing: contains videos of a teacher standing or walking. A standing action facing the students suggests a teacher's openness.

- Writing: contains videos of a teacher writing on a board. These videos are based on hand movement since we cannot see what is being written on board because the quality of the video is not good enough. A writing action suggests the engagement of the students using a visible medium.

- Pointing: contains videos of a teacher pointing to a board. A pointing action suggests the interaction.

- Talking: categorized videos of a teacher attending students. In all videos, students are standing close to the teacher. Talking also suggests the interaction, delivery and engagement.

- Cleaning: contains videos of a teacher cleaning the board with a duster or hand.

- Delivering Presentation, Teacher Standing: contains videos of the teacher standing while the projector displays slides. It is self-explanatory.

- Presentation, Writing: contains videos of the teacher writing on a board while the projector displays slides. It is self-explanatory as it suggests teacher's engagement.

- Delivering Presentation, Pointing on Board: This class contains videos of the teacher pointing to the board while the projector shows slides.

- Presentation, Talking: contains videos of a teacher and a student talking while the projector displays slides. It is self-explanatory as it suggests teacher's delivery.

- Presentation, Cleaning: contains videos of a teacher cleaning the board while the projector shows slides.

- Writing, Talking: contains videos of a teacher writing on a board while students standing close to the teacher and interacting with him. A writing action suggests the engagement of the students using a visible medium.

Sample images from selected videos from the dataset are depicted in Fig 1.

## 3.2 Background

Inductive bias [47] assorted various compositions of ANN and their utilities have been wedded with certain problem domains. A challenge, however, is the modeling of event-based

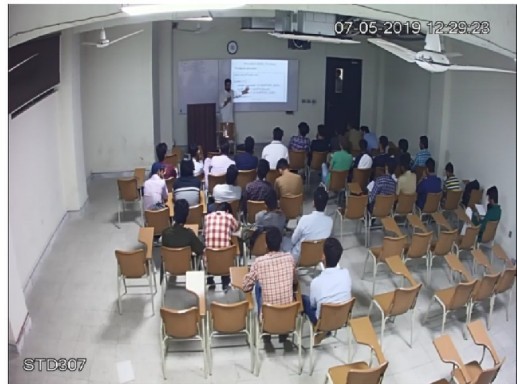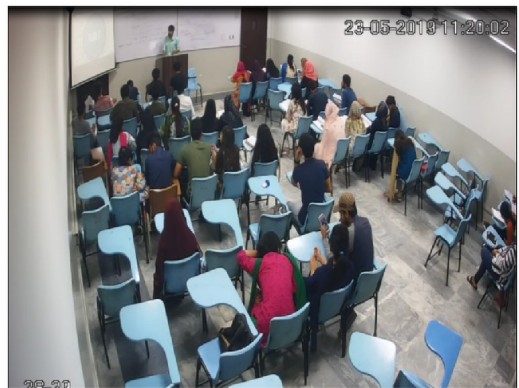

**Fig 1. In this figure two sample images from the dataset are shown.**

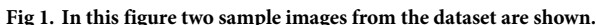

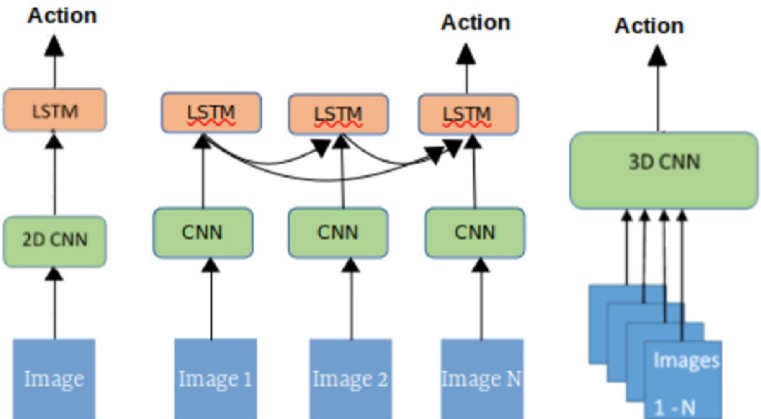

**Fig 2.** This figure depicts architectures of three reported DNN models, side by side, (a) CNN and LSTM implemented using Time distributed layer. An image frame from the video is given as input at one time step and action is predicted in the end. (b) Conv2DLSTM: image frames at each time step is presented to a the network and the features from all the images are used to action prediction. (c) 3DCNN: all images are concatenated and presented to the network for an action prediction.

segmentation. An event-based segmentation problem is often mapped on a clock-based segmentation. A motivation for exploring models with an inductive bias for our framework, this study probes multiple DNN models including 3D CNN, Conv2DLSTM, and time distributed with 2D CNN and LSTM. Fig 2 shows block diagrams of the investigated models. Details of the selected models are given in the following sections.

**3.2.1 Conv2D LSTM DNN.** A video data set is spatio-temporal and a sequential learning algorithm can be employed to explore the relationship of the frame. A recurrent neural network (RNN) exhibits an inductive bias toward time data, thus corresponds to equivariance in the time property of a learning algorithm. An LSTM unit in RNN is a complex activation function that carries a sequence of useful information in the form of representations from previous time steps and integrates it with the current time step for prediction. The representations, cell states, are trimmed and decorated with the activation from the hidden units at the previous time step and inputs at the current time step with the help of forgetting and input activation functions. The output activation of the current time step is computed using the hidden activation and cell state at the previous time step, and input at the current time step. An LSTM unit is depicted in Fig 3 and the following are the equations for the computation of an RNN with LSTM units:

$$C_t' = tanh(W_c[h_{t-1}, X_t]) + b_c$$
$$i_t = \sigma(W_i[h_{t-1}, X_t] + b_i)$$
$$o_t = \sigma(W_o[h_{t-1}, X_t] + b_o)$$
$$f_t = \sigma(W_f[h_{t-1}, X_t] + b_f)$$
$$C_t = i_t * C_t' + f_t * C_{t-1}$$
$$h_t = o_t * tanh(C_t)$$

(1)

where $W$ are wight matrix, $i$, $o$, $f$ and $C$ are input, output, forget gates and cell states, $t$ is the

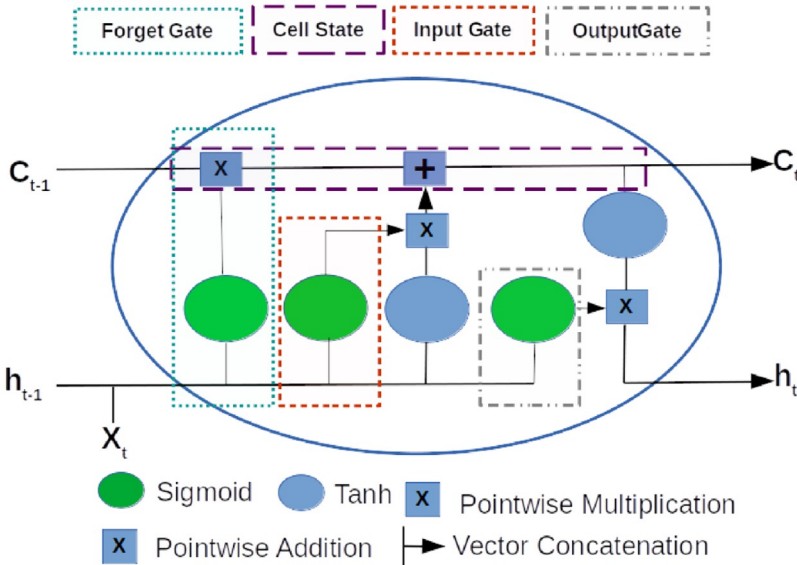

**Fig 3. This figure depicts a LSTM node of a RNN.** $C_{t-1}$ is cell state at previous frame and $C^t$ is the cell state at processing image frame. $h_{t-1}$ and $h_t$ are hidden states activation at previous and current time steps respectively. The patterned boxes depict the forget ($f_t$), input ($i_t$) and output ($o_t$) gates, respectively at current time step.

time and $b$ is for bias. A convolution operation equation for a single location $(i, j)$ is given as:

$$r_{ij} = \phi\left(\sum_{m=0}^{M-1}\sum_{n=0}^{N-1}\sum_{k} w_{m,n,k} v_{i+m-1, j+n-1, k}\right) \quad (2)$$

where $M$, $N$ are dimensions of the kernel, $k$ is the filter size and $\phi$ is the activation function.

However, the image frames at each time step are spatially diverse and pixels dimensions are too many inputs to be passed to LSTM layers in an RNN. A CNN, on the other hand, works well with the image data by extracting the localized features, and it exhibits group equivariance over space. Conv2D LSTM [48] is a combination of Convolution 2D of a CNN and LSTM unit (Fig 3) of RNN which reduces the implicit redundancy of pixels in an RNN. In a Convolution 2D layer, the weight is multiplied by the input similarly as it is done in the convolution operation. The composite LSTM units are addendum and the gates as well as input and cell states are localized within the neighborhood as shown in Fig 4. Following are the changed equations for Conv2DLSTM:

$$C'_t = tanh(W_{hc} \star h_{t-1} + W_{xc} \star X_t) + b_c$$

$$i_t = \sigma(W_{hi} \star h_{t-1} + W_{xi} \star X_t + W_{ci} C_{t-1} + b_i)$$

$$o_t = \sigma(W_{ho} \star h_{t-1} + W_{xo} \star X_t + W_{co} C_{t-1} + b_o)$$

$$f_t = \sigma(W_{hf} \star h_{t-1} + W_{xf} \star X_t + W_{cf} C_{t-1} + b_f) \quad (3)$$

$$C_t = i_t * C'_t + f_t * C_{t-1}$$

$$h_t = o_t * tanh(C_t)$$

where $\star$ is a convolution operation and the rest of the operations are the same as given in Eq 1.

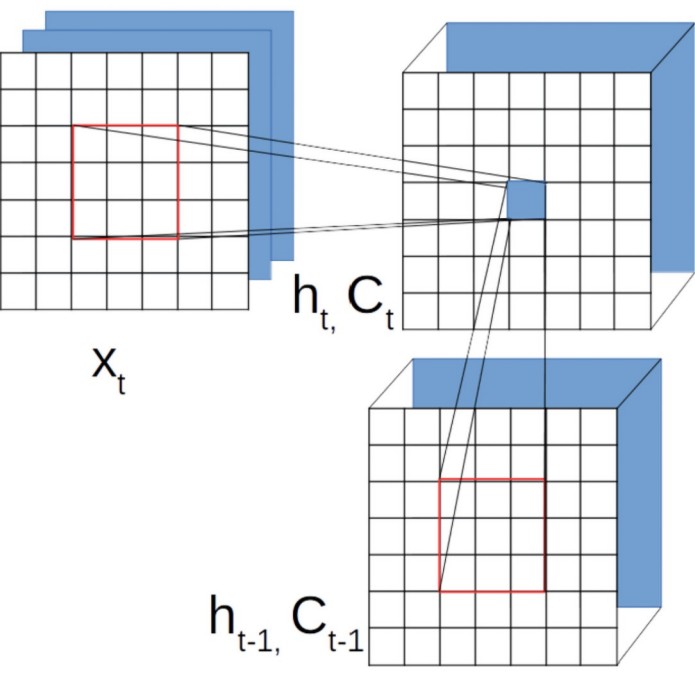

**Fig 4. This figure illustrates convolutional LSTM.** $X_t$ depicts the image and the red block in the middle encloses the local representation. $h_{t-1}$ is the hidden state of the processing of the previous image frame and $C_{t-1}$ is the cell state. $h_t$ and $C_t$ are the hidden and cell states respectively computed from $(X_t)$, $h_{t-1}$ and $C_{t-1}$. A red box shows the neighboring states that contribute to computation.

The input of Conv2DLSTM is a tensor with 5 dimensions, where the first two channels are samples and frames in the videos and the last three dimensions are the height, the width, and the channels in each frame.

**3.2.2 3D CNNs.** CNN is an established supervised learning technique for image data that works with individual spatial data. Adding a new human perceptible dimension of input data say time, a new computational dimension is a potent extension in the hard inductive bias. 3DCNN [49] is a composition of convolutional neural networks where a combination of subsequent data is used as an input tensor with an assumption that the tensor encompasses information for a single prediction. An illustration of 3DCNN is depicted in Fig 5. In comparison to the Eq 2 of 2D convolution, 3D convolution of a location $(i, j, l)$ (where $l$ is a time dimension) is given as:

$$r_{ijl} = \phi\left(\sum_{m=0}^{M-1}\sum_{n=0}^{N-1}\sum_{o=0}^{O-1}\sum_{k} w_{m,n,o,k} v_{i+m-1,j+n-1,l+o-1,k}\right) \tag{4}$$

where $M$, $N$, $O$ are dimensions of the kernel, $k$ is the filter size and $\phi$ is the activation function.

## 3.3 Proposed study

In this study, the architectures (section 3.2.1, Fig 5) possessing hard inductive bias are trained with the data prepared for teaching-methodology analysis. The inductive bias of a deep neural network can be improved for a specific domain and it has been experienced that the choice of hyper-parameters, post-processing of domain-specific data and initialization of the parameters (hyper-parameters and learn-able parameters) transcend the general models. The three architectures depicted in Fig 6 are trained with an image sequence $\{I_1, I_2, \ldots, I_N\}$ extracted from the

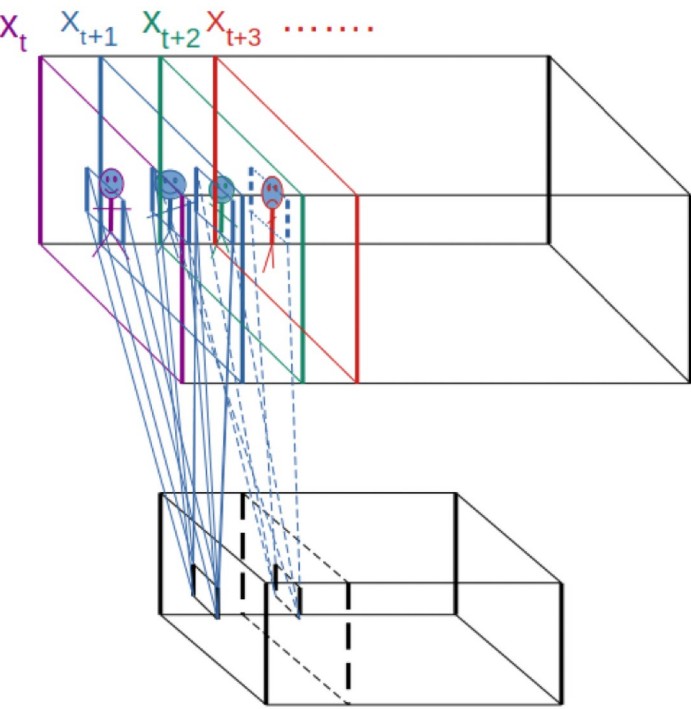

**Fig 5. This figure illustrates 3DCNN.** The rectangle on top depicts slices of video frames and each slice is an image that is color coded. Contiguous frames convolve with a 3D kernel and it is depicted with regular and dashed lines. The rectangle at the bottom is a depiction of the 3D convolution of image tensor with kernel tensor (it should not be confused with the shared filters).

video $V_a$ of action category $a$. In this section, we discuss the composition of the proposed networks. The selected hyper-parameters are discussed in section 4.

**3.3.1 ConvLSTM for teaching methodology analysis.** Fig 7a gives the detailed composition of the first model. The network has four layers Conv2DLSTM with interleaving sub-sampling layers. The input frame $I_t$ at time $t$ is clamped to the first layer and extracts the features

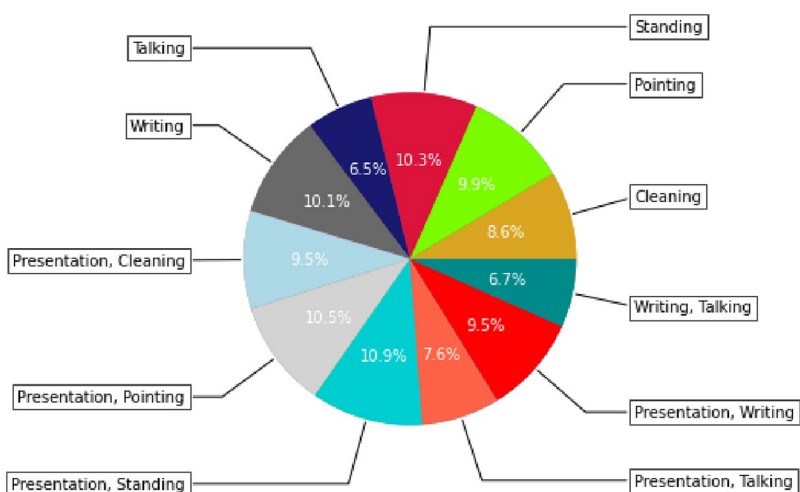

**Fig 6. The graph in this image depicts a distribution of classes in training data in a LOOCV sample.**

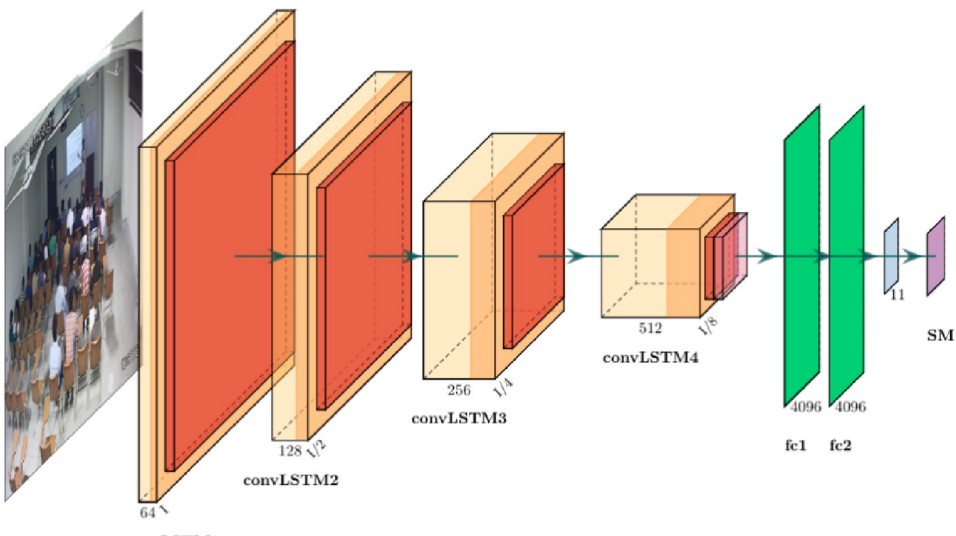

**(a)** ConvLSTM network composition.

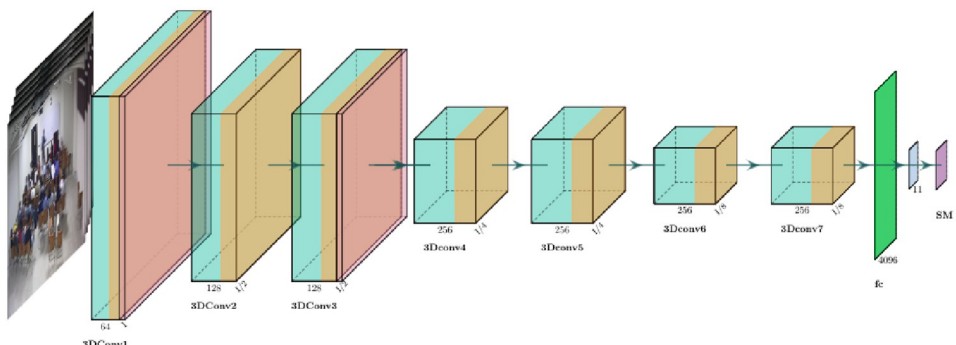

**(b)** 3DCNN network composition.

**Fig 7.** (a) shows the composition of a Conv2DLSTM network. The receptive field is adjusted using sub-sampling and strides (convLSTM is a Conv2DLSTM layer, fc is fully connected and SM is softmax layer). (b) shows the composition of a 3DCNN. The receptive fields of the subsequent layers are adjusted using the strides only (3DConv is a 3DCNN layer). The numbers at the bottom of each shape give the number of filters in that layer. I/x term given at the bottom corner of each layer shows the reduced size of the features after application of subsampling layer by 1/x times of the size (H and W seprately) of image I.

from the images using 3x3 convolutional kernels in convolutional layers, whereas the LSTM units keep the context from subsequent images $\{I_1, \ldots, I_t\}$ of the whole video. The encoded features from the final sub-sampling layer are forwarded to a feed forward network (FC3) with three fully connected layers including an output layer followed by a softmax layer. A dropout layer with 50% probability is added after the final sub-sampling layer. The network predicts an action at the end of the whole sequence of image frames in a video at test time. The number of filters in each layer are written under a shape in Fig 7a. The size of the receptive field of each convLSTM layer is calculated based on the input images height and width (HxW) and kernerl sizes of the previous layer.

In this study, the convolution and LSTM are tested with two different compositions: Conv2DLSTM recurrent network builds the context from the cell states and the hidden states

                                              

generated by LSTM layers using all the subsequent image frames $\{I_1, \ldots, I_t\}$ in the sequence for a current input $I_t$. Whereas in the convolution time distributed layers, the network only builds context from the LSTM outputs of the last image frame $I_{t-1}$ for the input $I_t$. Further-more, output of the Conv2DLSTM layer can be either 5 or 4 dimensional based on the implementation strategy for input dimension that can have single input for the whole video or single input for each image in the temporal sequence.

**3.3.2 3DCNN for teaching methodology analysis.** Fig 7b gives the detailed composition of the 3DCNN model used for a teacher's action classification. A video is a sequence of stacked images that can be represented as a tensor, where width and height are two dimensions in the tensor and time is the third dimension. The action recognition spans over the contiguous image frames and it can be visualized as a certain pattern in a tensor. The proposed architecture of a 3DCNN is composed of seven 3D convolution layers with two dropout layers with 50% probability after the first and third 3D convolutional layers. As shown in Fig 2, the whole sequence of images $\{I_1, \ldots, I_N\}$ in a video is concatenated and clamped to the input layer. The convolutional layers extract features and learn the association of features across the sequence of image frames. The features from the final layer are passed to a fully connected layer and a final output layer predicts an action $a$ using a softmax layer.

The input of Conv3D is five-dimensional; the 1st dimension is the number of samples, the 2nd dimension is depth, the 3rd is width, the 4th is height, and the 5th is the number of channels. In the video dataset usage with 3DCNN, samples are numbers of video clips in a batch, and depth is the number of frames in a video clip.

## 4 Results, discussion and limitations

Several experiments are performed with the collected video dataset discussed in section 3.1 using the Deep Learning models discussed in section 3.2. Fig 7a and 7b give details of the compositions of DNNs employed for experimentation in this study. The activation units in convolution layers are ReLU, and for optimization, Adam stochastic optimizer is used with following hyper parameters: $\alpha = 0.001$, $\beta_1 = 0.9$, $\beta_2 = 0.999$ and $\epsilon = 10e^{-8}$. Here $\alpha$ is the learning rate, $\beta_1$ is the exponential decay rate for the first-moment estimate, $\beta_2$ is the exponential decay rate for the second-moment estimate and $\epsilon$ is used to avoid division by zero in the implementation in case of zero gradient [50]. All experiments use multi-class cross entropy loss function and a batch size of one.

The training and testing datasets are drawn out from the labeled video clips. The proportion of video clips in each category are depicted in histograms in Figs 6 and 8. Before feeding the dataset into the Deep Learning model, it is normalized by changing the channel information and pixel values. Initial values for pixels were between 0 and 255 but during preprocessing these were normalized between 0 and 1. We also reduced the size of the frames from 704x576 to 360x240.

The dataset is also augmented to increase the number of available samples and add a data inductive bias. Data augmentation includes: image rotation with an angle of ±5˚ (shocks/jerks/jolt invariance), flipping horizontally (viewpoint invariance), changing brightness (illuminance invariance), and zooming in or zooming out (scale invariance). The advantage of using data augmentation is that our training data become more diverse, and an increase in the number of training samples also improves the performance of deep neural networks with better generalization. The flipping and the brightness augmentation operations are performed on a complete labeled clip, whereas the other operations are performed on image frames within a clip. For the former, a clip is selected at random with a probability of 0.2. The image rotation is performed on an image frame randomly selected from a clip with a probability of 0.1 and the

                                       

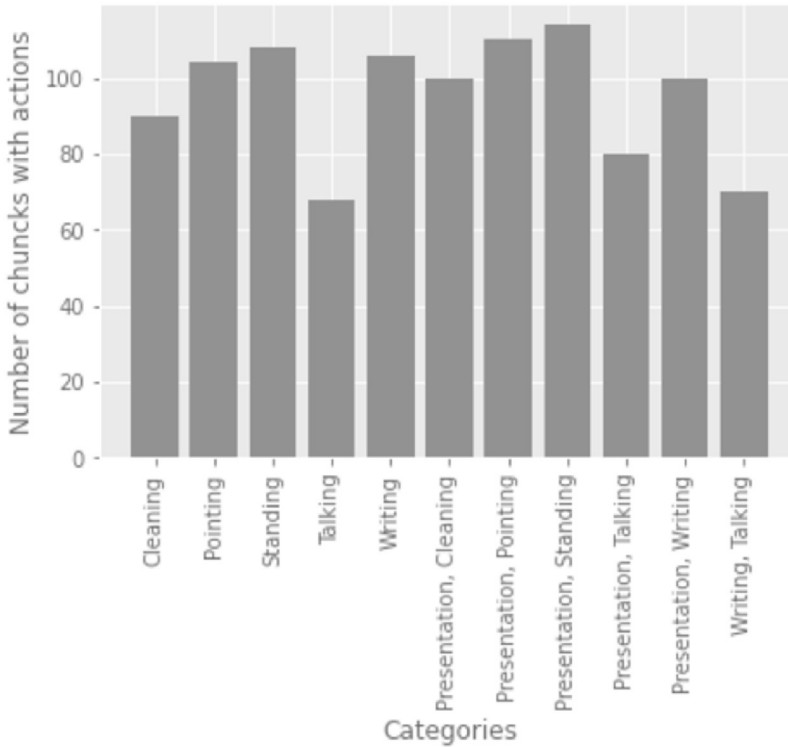

**Fig 8. The graph in this image depicts a distribution of classes in testing data in a LOOCV sample.**

angle sign is selected with a probability of 0.5. The rotation is performed on two consecutive frames in a selected angle and the opposite rotation is performed in the next four frames and then the selected rotation is performed in two consecutive frames. This imitates a complete motion of the camera in case of shock or jolt. The zooming and zooming out are also performed on a randomly selected image frame with a probability of 0.1 from a clip. The zooming operations are performed with interpolation and in the case of zooming out the operation, the borders are filled with mirror padding.

DNN models of Conv2DLSTM and Convolution with time distributed layers are trained with one frame at a time, and 3DCNN is trained with all frames in a clip (as mentioned in section 3.1)) stacked vertically. A leave-one-out cross-validation (LOOCV) technique is used for evaluations. The chunks for the video selected for testing are not included in training data to achieve a true out-of-distribution generalization. Moreover, the chunks from the different videos are interleaved and shuffled for training in addition to the data augmentation disscussed in previous paragraph. Accumulated figures of validation accuracy results with the tested models are presented in Table 1. It is observed that the 3DCNN network is effective with the videos dataset since their composite layers work well with spatio-temporal data. Although, time-

**Table 1. This table presents the accuracy of the tests averaged over nine runs with LOOCV cross validation techniques.**

| Models | Average accuracy ± SD |
| --- | --- |
| TD 2D CNN + LSTM | 91.00% ±2 |
| Conv2DLSTM | 91.00% ±1.5 |
| 3D CNN | 94.00% ±3 |

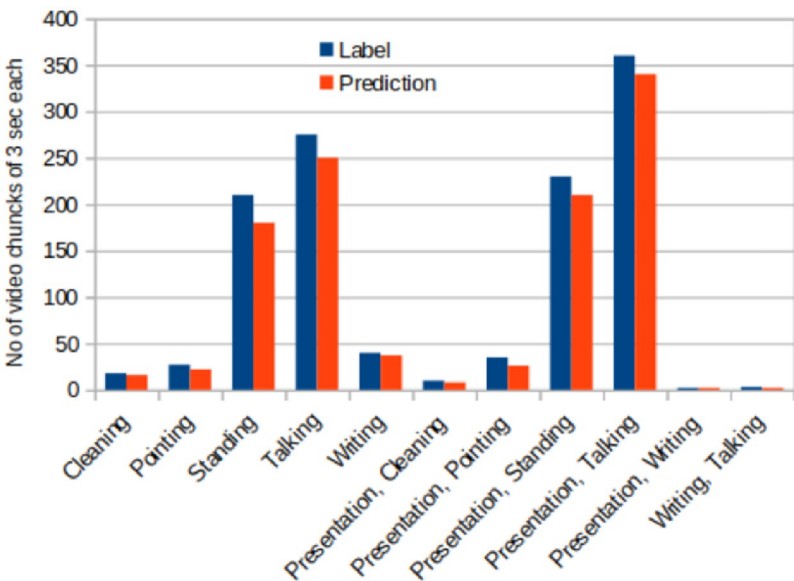

**Fig 9. This chart depicts statistics of the classified actions in a complete lecture video of one hour.** The results are generated with Conv2DLSTM. These results can be correlated with the teaching standards.

distributed layers with CNN, LSTM and Conv2DLSTM have hard inductive bias for retaining the long sequential association of the data, the quantitative results suggest the complex composition of 3DCNN layers adds benefits in this particular problem. It extracts more valuable features and keeps the right temporal associations that are needed for accurate predictions with a small and noisy sequential images dataset.

In this study, we also use a complete lecture video to generate summary statistics. First, the video is split into 3 seconds chunks. Each chunk is separately fed into the model to predict the action performed by a teacher. Finally, the results are generated by measuring the number of classified chunks. In this experiment, we were able to get 90% accuracy for a single lecture video of 1 hour. In Fig 9, we can see specific actions that were performed during the lecture. Y-axis shows the number of three seconds clips in which that action was performed. A confusion matrix of results of the video is presented in Fig 10, and it shows which actions are confused with other actions.

According to lecturing guidelines published online by the University of Waterloo (https://uwaterloo.ca/centre-for-teaching-excellence/teaching-resources/teaching-tips/lecturing-and-presenting/delivery/lecturing-effectively-university. Latest visit(11th May 2021) [51, 52], it is recommended that a teacher should turn her face towards the students, and she should not spend most of her time just writing on board or looking at the slides. Communicating with students is considered the most vital part of a lecture. So, based on this observation, interesting actions of a teacher according to our labeled categorizes are: talking to a student, pointing to the board, and standing while facing students. A lecture in which a teacher spends a substantial amount of time in these interesting categories of actions along with presentation and writing on a board can be considered an effective teaching session.

## 4.1 Discussion

In this section, we reflect on the challenges peculiar to teaching methodology analysis using CCTV videos. A significant list is the poor quality of CCTV videos (Fig 11), imprecise class

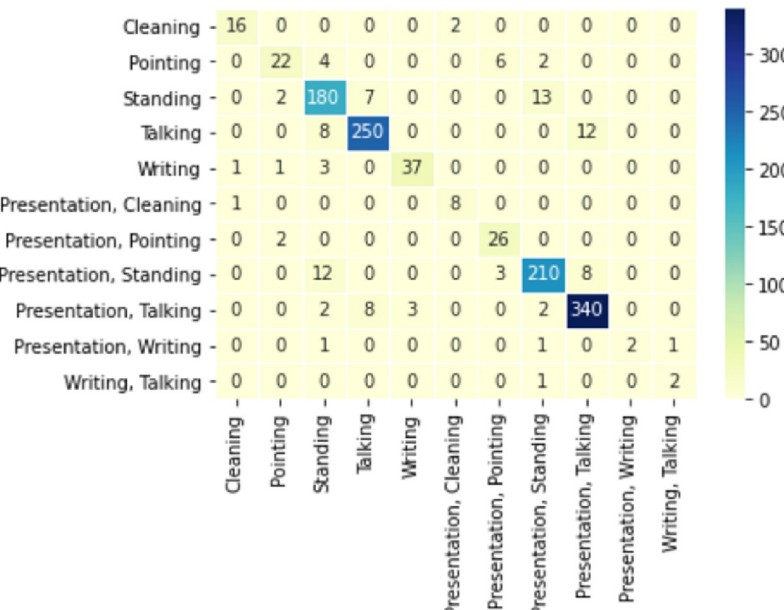

**Fig 10. This image depicts a confusion matrix of the stats presented in Fig 9.**

labels, and high inter-class similarity. The videos used in this study are CCTV videos and lack important details due to the point of view and quality of CCTV cameras. Often, it is not discernible with a naked eye if a teacher is holding a marker/chalk. Moreover, in multiple videos, the whiteboard is not visible and the teacher's writing on the board becomes illegible, as depicted in Fig 11.

Another limitation is overlapping actions in consecutive video chunks of the collected dataset. Some of the intermediate actions do not have any precise labels such as teacher walking, teacher waving hand, or teacher standing still. Some video clips have very brief recognizable actions either at the beginning or the ending of the video, such as a three-second video that has

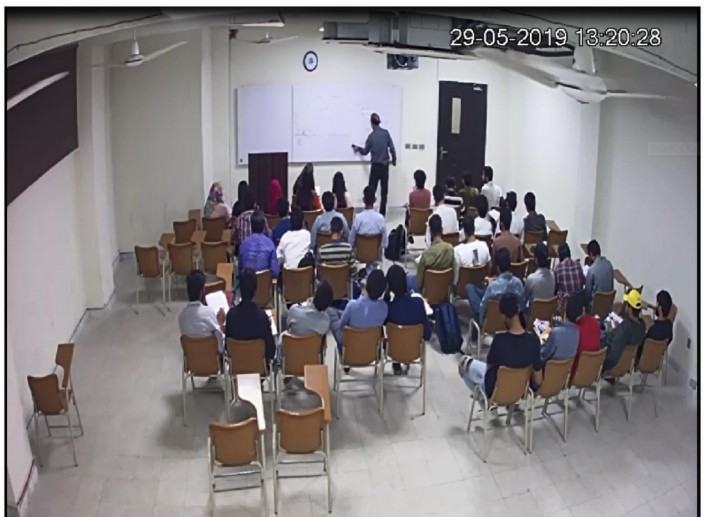

**Fig 11. Poor quality video example as writing on board is not visible.**

**Table 2. Comparison with IAVID-1 dataset.**

| Our Dataset | IAVID-1 Dataset |
|---|---|
| 1050 Training + Testing Videos | 100 Training + Testing Videos |
| 11 Total Actions | 8 Total Actions |
| 6 teachers | 12 actors, Each actor perform 8 different actions |
| Videos contain Front view from top left or top right (Whole Class View) | Videos contain front view (Teacher focused view) |
| Mixed actions, at most 2 type of action per video | Single action per video |
| Highest accuracy was 94% | Highest accuracy was 82% |
| 704x576 scale down to 360x240 | 1088x1920 |

less than one second of writing on board. Since it is less than a second, so we chose not to classify these as the action being performed. In some videos, less than three seconds of a certain action is performed, which is divided into two different video clips because the videos are split evenly at three seconds mark. This makes it hard to get actual statistics. To overcome this challenge, a strategy of frame dropout was adopted but it did not improve the results, instead, it was exacerbated. A detailed analysis is needed, however, to find that if it is an inherent bias of the dataset.

Another challenge we faced in this study is high inter-class similarity, such as pointing on the board and writing on the board are almost similar actions. So, it becomes difficult to classify these as separate classes. This problem aggravates because of the poor quality of the videos. Another case is a student and a teacher talking, i.e., sometimes a few students stand close to the teacher or pass by the teacher. It is not easy to classify when they are talking since our model does not include sound input. Another example is delivering a presentation where sometimes a teacher does not move much or points towards the presentation so it is also hard to recognize.

It is important to discuss here the utility of background segmentation and region demarcation in a video clip for training and testing. First, background segmentation is not used in this study because modern cameras have enhanced functionality of following a person. Moreover, PTZ cameras are frequently used in video recording these days. The region demarcation is not used to restrict the focus on the teacher as communication with students and their response is significant for some classes. However, these two preprocessing techniques can be tested in the future for certain objectives and can help to improve results.

We compared our dataset with the IAVID-1 dataset. This dataset is similar to our dataset in terms of output classes and the purpose of this dataset aligns with our study, but our dataset is more generalized which is evident from facts in Table 2. The IAVID-1 dataset is very simple in terms of the number of videos per output class, and the view of videos focused on the teacher only. This may not be considered as a true representation of a complete classroom environment, since communication with students is an essential component of the lecture. However, the two datasets can be merged and used for further analysis and evaluations.

## 5 Conclusion and future work

In this study, automatic content recognition (ACR) is experimented with real-world classroom video data for analysis of teaching methods. It facilitates monitoring, self-reflection, and adaptation of an improved teaching style. The actions of a teacher are categorized and labeled in classroom videos, and a comprehensive dataset of labeled video clips is generated. The dataset is used to train Deep Learning action recognition models. The proposed deep neural

network-based model generates valuable statistics which are effectively tested on 1000 video clips for quantitative assessments. Furthermore, the use of ACR techniques for the analysis of online lecture videos is a pertinent study and becomes indispensable with online teaching due to the Covid-19 pandemic which caused a forced shutdown of educational campuses around the world. The proposed models can be extended with experiments on online sessions with students' action recognition and a comprehensive analysis of the effectiveness of online teaching sessions.

## Author Contributions

**Conceptualization:** Muhammad Aasim Rafique, Malik Tahir Hassan, Moongu Jeon.

**Data curation:** Muhammad Aasim Rafique, Faheem Khaskheli.

**Formal analysis:** Faheem Khaskheli, Malik Tahir Hassan, Sheraz Naseer, Moongu Jeon.

**Funding acquisition:** Moongu Jeon.

**Investigation:** Muhammad Aasim Rafique, Sheraz Naseer.

**Methodology:** Muhammad Aasim Rafique, Faheem Khaskheli, Malik Tahir Hassan, Sheraz Naseer.

**Project administration:** Moongu Jeon.

**Supervision:** Malik Tahir Hassan, Moongu Jeon.

**Visualization:** Muhammad Aasim Rafique, Faheem Khaskheli.

**Writing – original draft:** Muhammad Aasim Rafique, Faheem Khaskheli.

**Writing – review & editing:** Malik Tahir Hassan, Sheraz Naseer, Moongu Jeon.

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
