## [Decision Letter · Decision Letter 0]

20 Oct 2021

PONE-D-21-18820Employing Automatic Content Recognition for Teaching Methodology Analysis in Classroom VideosPLOS ONE

Dear Dr. Jeon,

Thank you for submitting your manuscript to PLOS ONE. After careful consideration, we feel that it has merit but does not fully meet PLOS ONE’s publication criteria as it currently stands. Therefore, we invite you to submit a revised version of the manuscript that addresses the points raised during the review process.

We look forward to receiving your revised manuscript.

Kind regards,

Felix Albu, Ph.D.

Academic Editor

PLOS ONE

Journal Requirements:

2. Please note that PLOS ONE does not allow footnotes, so please include all text in the footnotes in the main text. Please also include the name of the Ethics committee which approved the study, and clarify how participants provided consent.

“We would like to thank the University of Management and Technology for providing the 430 CCTV camera lecture videos. This work was supported by a grant from the Institute of 431 Information & Communications Technology Planning & Evaluation (IITP) funded by 432 the Korean government (MSIT) (No. 2014-3-00077, AI National Strategy Project).”

“This work was supported by Institute of Information & communications Technology Planning & Evaluation (IITP) grant funded by the Korea government(MSIT) (No.2014-3-00077, AI National Strategy Project).”

4. We note that Figure 1 includes an image of a [patient / participant / in the study]. As per the PLOS ONE policy (http://journals.plos.org/plosone/s/submission-guidelines#loc-human-subjects-research) on papers that include identifying, or potentially identifying, information, the individual(s) or parent(s)/guardian(s) must be informed of the terms of the PLOS open-access (CC-BY) license and provide specific permission for publication of these details under the terms of this license. Please download the Consent Form for Publication in a PLOS Journal (http://journals.plos.org/plosone/s/file?id=8ce6/plos-consent-form-english.pdf). The signed consent form should not be submitted with the manuscript, but should be securely filed in the individual's case notes. Please amend the methods section and ethics statement of the manuscript to explicitly state that the patient/participant has provided consent for publication: “The individual in this manuscript has given written informed consent (as outlined in PLOS consent form) to publish these case details”.

Additional Editor Comments (if provided):

The authors have to address the comments of the reviewers.

Reviewers' comments:

Reviewer's Responses to Questions

**Comments to the Author**

1. Is the manuscript technically sound, and do the data support the conclusions?

Reviewer #1: Partly

Reviewer #2: Yes

2. Has the statistical analysis been performed appropriately and rigorously? 

Reviewer #1: Yes

Reviewer #2: No

3. Have the authors made all data underlying the findings in their manuscript fully available?

Reviewer #1: No

Reviewer #2: Yes

4. Is the manuscript presented in an intelligible fashion and written in standard English?

Reviewer #1: Yes

Reviewer #2: Yes

5. Review Comments to the Author

Reviewer #1: The article presents the importance of the existence of a module for recognizing human actions that can be used to improve the teaching act. Also, this includes a description of how the dataset was collected and three types of video classifiers.

I suggest the following improvements:

Section 2.1 should detail the preprocessing stage (what transforms are used for data preprocessing before training). It is specified that augmentation was used only in the results section.

Figure 2 can be improved because it is not explicit enough.

- it is not clear if the first architecture uses only the first and last image or the whole video;

- in the second architecture, it is not clear if only three frames;

- it is not clear how the prediction of the action is determined, starting from the output of the LSTMs;

- it is not clear whether a convolutional layer is used for each image or all images are passed through the same layer.

Section 3 contains general information about various types of neural networks but does not include a clear presentation of the proposed methods and the pipeline used to train them. This section could be renamed and used as Background and introduced a new section with the presentation of the proposed methods (containing a presentation specific to the problem solved).

In section 4, it is not specified how the ADAM optimizer was chosen, what value was used for the learning rate and if other variants were tested.

Figure 6 could also include the percentages for each type of action. The size used for the batch is not specified. The error function used is not specified.

It would be useful if Table 2 turned into a figure. It would be easier to understand the proposed model.

The authors should point out the contributions they bring in the context of human activity recognition or video classification compared to the other approaches presented in the state of the art section.

A comparison should be made between the results obtained by these three methods presented.

Reviewer #2: Comments to the Author:

This paper has proposed a teaching methodology analysis in classroom videos. The work of this paper is practical. However, I think there are some revisions for the manuscript. The comments are list as follow.

(1) The author needs to analyze the difference of effect for each label.

(2) The author does not reflect the novelty of the method. The method used is the existing method. Whether it is advanced in this field needs to be compared with the existing advanced method in this field.

(3) The latest references cited by the author contain too few articles in this direction.

(4) The first appearance abbreviation should show all words, such as CCTV in line 71 and IDTs in line 77.

6. PLOS authors have the option to publish the peer review history of their article (what does this mean?). If published, this will include your full peer review and any attached files.

Reviewer #1: No

Reviewer #2: No

---

## [Author Response · Author response to Decision Letter 0]

9 Nov 2021

We are grateful to the respected editor, the editorial team and erudite reviewers for extending this opportunity to improve our manuscript. We are thankful for the insightful comments and valuable suggestions by the erudite reviewers. It encouraged us to improve the proposed study and the quality of the manuscript. We diligently deliberated the suggestions, supplemented the proposed study and ameliorated the manuscript accordingly. We address the journal requirements guidelines and comments of each respected reviewer separately in the following:

Journal Requirements:

Response: The revised manuscript complies the PLOS ONE’s style requirements.

2. Please note that PLOS ONE does not allow footnotes, so please include all text in the footnotes in the main text. Please also include the name of the Ethics committee which approved the study, and clarify how participants provided consent.

Response: The footnotes are accommodated in the text and a letter including the names of members of ethics committee with details about consent is provided as supplementary document. 

“We would like to thank the University of Management and Technology for providing the 430 CCTV camera lecture videos. This work was supported by a grant from the Institute of 431 Information & Communications Technology Planning & Evaluation (IITP) funded by 432 the Korean government (MSIT) (No. 2014-3-00077, AI National Strategy Project).”

“This work was supported by Institute of Information & communications Technology Planning & Evaluation (IITP) grant funded by the Korea government (MSIT) (No.2014-3-00077, AI National Strategy Project).”

Response: A revised funding and acknowledgment statement is provided in cover letter. 

4. We note that Figure 1 includes an image of a [patient / participant / in the study]. As per the PLOS ONE policy (http://journals.plos.org/plosone/s/submission-guidelines#loc-human-subjects-research) on papers that include identifying, or potentially identifying, information, the individual(s) or parent(s)/guardian(s) must be informed of the terms of the PLOS open-access (CC-BY) license and provide specific permission for publication of these details under the terms of this license. Please download the Consent Form for Publication in a PLOS Journal (http://journals.plos.org/plosone/s/file?id=8ce6/plos-consent-form-english.pdf). The signed consent form should not be submitted with the manuscript, but should be securely filed in the individual's case notes. Please amend the methods section and ethics statement of the manuscript to explicitly state that the patient/participant has provided consent for publication: “The individual in this manuscript has given written informed consent (as outlined in PLOS consent form) to publish these case details”.

Response: A letter from the ethics committee providing the consent of the participants is provided as a supplementary document. 

Response: The requirement is accommodated in the revised manuscript. 

Additional Editor Comments (if provided):

The authors have to address the comments of the reviewers.

Reviewers' comments:

Reviewer's Responses to Questions

Comments to the Author

1. Is the manuscript technically sound, and do the data support the conclusions?

Reviewer #1: Partly

Reviewer #2: Yes

Response: We are grateful for the acknowledgment by the reviewers and improved the deficiencies with the help of suggested changes and feedback. 

2. Has the statistical analysis been performed appropriately and rigorously?

Reviewer #1: Yes

Reviewer #2: No

Response: We acknowledge the insightful feedback provided by the reviewers and revised the manuscript anticipating the astute observations by the reviewers. 

3. Have the authors made all data underlying the findings in their manuscript fully available?

Reviewer #1: No

Reviewer #2: Yes

Response: We are grateful for the acknowledgment by the reviewers.

4. Is the manuscript presented in an intelligible fashion and written in standard English?

Reviewer #1: Yes

Reviewer #2: Yes

Response: We are grateful for the acknowledgment by the reviewers.

5. Review Comments to the Author

Reviewer #1: The article presents the importance of the existence of a module for recognizing human actions that can be used to improve the teaching act. Also, this includes a description of how the dataset was collected and three types of video classifiers.

I suggest the following improvements:

Section 2.1 should detail the preprocessing stage (what transforms are used for data preprocessing before training). It is specified that augmentation was used only in the results section.

Response: Section 2.1 is updated with the pre-processing and data augmentation information. Moreover, the details are appended in section 3 (Results, Discussion and limitation). Briefly, this study uses horizontal flip, zoom-in, zoom-out, change of brightness, image rotation and image blur for data augmentation. 

Figure 2 can be improved because it is not explicit enough.

Response: Figure 2 has been updated in the manuscript and the caption provides additional information to remove the ambiguities. The description below is also updated section 2 of the manuscript for better readability and understanding. 

- it is not clear if the first architecture uses only the first and last image or the whole video;

Response: It takes one image at one time step ‘t’ and predicts an action after t_N (N number of images). 

- in the second architecture, it is not clear if only three frames;

Response: It takes one image at one time step ‘t’ and predicts an action after t_N (N number of images). However, in implementation we can pass all N images at the same time. 

- it is not clear how the prediction of the action is determined, starting from the output of the LSTMs;

Response: The action is predicted at the end of a sequence. 

- it is not clear whether a convolutional layer is used for each image or all images are passed through the same layer.

Response: CNN is used for each image in architectures 1 and 2. While in architecture 3 all the images are clamped at once to the 3DCNN. 

Section 3 contains general information about various types of neural networks but does not include a clear presentation of the proposed methods and the pipeline used to train them. This section could be renamed and used as Background and introduced a new section with the presentation of the proposed methods (containing a presentation specific to the problem solved).

Response: As per the suggestion, a new subsection has been added to the manuscript with precise details of the proposed techniques relevant to the selected problem. 

In section 4, it is not specified how the ADAM optimizer was chosen, what value was used for the learning rate and if other variants were tested.

Response: The details for Adam optimizer have been added in section 3 (Results, Discussion and Limitations). 

Figure 6 could also include the percentages for each type of action. The size used for the batch is not specified. The error function used is not specified.

Response: As per suggestion Figure 6 is updated and the details of batch size and the error /loss function information are added in section 3 (Results, Discussion and Limitations). 

It would be useful if Table 2 turned into a figure. It would be easier to understand the proposed model.

Response: As per suggestion Table 1 and Table 2 are transformed into figures (Figure 10 (a) and (b)). 

The authors should point out the contributions they bring in the context of human activity recognition or video classification compared to the other approaches presented in the state-of-the-art section.

Response: The introduction section has been updated with elaborated contributions. 

A comparison should be made between the results obtained by these three methods presented.

Response: The comparison is presented in discussion section and updated in Table 3. 

Reviewer #2: Comments to the Author:

This paper has proposed a teaching methodology analysis in classroom videos. The work of this paper is practical. However, I think there are some revisions for the manuscript. The comments are list as follow.

(1) The author needs to analyze the difference of effect for each label.

Response: It is added to the section 2.1 of the manuscript. 

(2) The author does not reflect the novelty of the method. The method used is the existing method. Whether it is advanced in this field needs to be compared with the existing advanced method in this field.

Response: The introduction section is updated with elaborated contributions and novelty of the idea in this study.

(3) The latest references cited by the author contain too few articles in this direction.

Response: The reference section is updated with recent relevant articles. 

(4) The first appearance abbreviation should show all words, such as CCTV in line 71 and IDTs in line 77.

Response: The manuscript is carefully revised and all words are used for the first use of an abbreviation.

---

## [Decision Letter · Decision Letter 1]

30 Dec 2021

PONE-D-21-18820R1Employing Automatic Content Recognition for Teaching Methodology Analysis in Classroom VideosPLOS ONE

Dear Dr. Jeon,

Thank you for submitting your manuscript to PLOS ONE. After careful consideration, we feel that it has merit but does not fully meet PLOS ONE’s publication criteria as it currently stands. Therefore, we invite you to submit a revised version of the manuscript that addresses the points raised during the review process.

We look forward to receiving your revised manuscript.

Kind regards,

Felix Albu, Ph.D.

Academic Editor

PLOS ONE

Journal Requirements:

Additional Editor Comments (if provided):

The comments of the reviewers should be better addressed.

Reviewers' comments:

Reviewer's Responses to Questions

**Comments to the Author**

1. If the authors have adequately addressed your comments raised in a previous round of review and you feel that this manuscript is now acceptable for publication, you may indicate that here to bypass the “Comments to the Author” section, enter your conflict of interest statement in the “Confidential to Editor” section, and submit your "Accept" recommendation.

Reviewer #1: (No Response)

Reviewer #2: All comments have been addressed

2. Is the manuscript technically sound, and do the data support the conclusions?

Reviewer #1: Partly

Reviewer #2: Yes

3. Has the statistical analysis been performed appropriately and rigorously? 

Reviewer #1: Yes

Reviewer #2: Yes

4. Have the authors made all data underlying the findings in their manuscript fully available?

Reviewer #1: Yes

Reviewer #2: Yes

5. Is the manuscript presented in an intelligible fashion and written in standard English?

Reviewer #1: Yes

Reviewer #2: Yes

6. Review Comments to the Author

Reviewer #1: Preprocessing information has been added, but it is too minimal. It was not specified with what probability these transformations were applied and what parameters are used for each transform (i.e. the scale used for zooming in / zooming out).

The authors did not specify all parameters for the neural networks (ConvLSTM / 3DCNN). For example, no parameters were specified for LSTMs such as the number of features in the hidden state h, the number of recurrent layers. The two proposed models should be better detailed. Some of this information existed in the previous table.

The authors provide general details: "The input of Conv3D is five-dimensional; the 1st dimension is the number of samples, the 2nd dimension is depth, the 3rd is width, the 4th is height, and the 5th is the number of channels. In the video dataset, samples are numbers of videos, and depth is the number of frames in a video". What is the significance of the number of videos in the context of the proposed solution?

The authors provided details for the optimization algorithm using notation symbols. It is necessary to explain what each notation represents. For example, alpha represents the learning rate, epsilon represents the term added to the denominator to improve numerical stability.

Reviewer #2: The author has made great modification to the structure of the paper. Recent references have been added as required. Meanwhile, the relative methods are compared. The paper has met the basic requirements of the journal. I don’t have any problems. I suggest to accept it.

7. PLOS authors have the option to publish the peer review history of their article (what does this mean?). If published, this will include your full peer review and any attached files.

Reviewer #1: No

Reviewer #2: No

---

## [Author Response · Author response to Decision Letter 1]

6 Jan 2022

6. Review Comments to the Author

Reviewer #1: Preprocessing information has been added, but it is too minimal. It was not specified with what probability these transformations were applied and what parameters are used for each transform (i.e. the scale used for zooming in / zooming out).

Response : As suggested by the reviewer the details of data augmentation operations used in this study are elaborated with selected parameters in section 3 (line 409-420 - page 22) (“Results, Discussion and Limitations”).

The authors did not specify all parameters for the neural networks (ConvLSTM / 3DCNN). For example, no parameters were specified for LSTMs such as the number of features in the hidden state h, the number of recurrent layers. The two proposed models should be better detailed. Some of this information existed in the previous table.

Response : As suggested by the reviewer the details from the mentioned tables are added to the images. The hyper-parameters of the neural networks are added in the text in images and elaborated in the captions. The details are pronounced in the text added earlier for clarity and better readability. (Figure 10 (image and caption), Section 2.3.1, line 359-362, page 22) 

The authors provide general details: "The input of Conv3D is five-dimensional; the 1st dimension is the number of samples, the 2nd dimension is depth, the 3rd is width, the 4th is height, and the 5th is the number of channels. In the video dataset, samples are numbers of videos, and depth is the number of frames in a video". What is the significance of the number of videos in the context of the proposed solution?

Response : The notified sentence is rephrased to clearly express the authors intention. The video clips (batch size) often play a critical role as an implicit regularizer, which in this study can not have much impact as a single clip is used in a batch. The frames in a clip form a volume and are assumed to serve the purpose of the batch, but the claims need further inquiry which can be pursued in future work. 

The authors provided details for the optimization algorithm using notation symbols. It is necessary to explain what each notation represents. For example, alpha represents the learning rate, epsilon represents the term added to the denominator to improve numerical stability.

Response : As suggested by the reviewer the descriptions are added to the manuscript in the Section 3 (line 393-395 – page 22) “ Results, Discussion and Limitations”. Moreover, following reference is added for a detailed reading.

“Kingma DP, Ba J. Adam: A Method for Stochastic Optimization. In: 3rd International Conference on Learning Representations, ICLR , San Diego, CA, USA, May 7-9, 2015.”

Reviewer #2: Comments to the Author:

The author has made great modification to the structure of the paper. Recent references have been added as required. Meanwhile, the relative methods are compared. The paper has met the basic requirements of the journal. I don’t have any problems. I suggest to accept it.

Response : We are grateful to the respected reviewer for providing us with valuable comments which helped us to improve the manuscript and prepare it for positive appraisal.

---

## [Decision Letter · Decision Letter 2]

20 Jan 2022

Employing Automatic Content Recognition for Teaching Methodology Analysis in Classroom Videos

PONE-D-21-18820R2

Dear Dr. Jeon,

We’re pleased to inform you that your manuscript has been judged scientifically suitable for publication and will be formally accepted for publication once it meets all outstanding technical requirements.

Kind regards,

Felix Albu, Ph.D.

Academic Editor

PLOS ONE

Additional Editor Comments (optional):

The decision is Accept.

Reviewers' comments:

Reviewer's Responses to Questions

**Comments to the Author**

1. If the authors have adequately addressed your comments raised in a previous round of review and you feel that this manuscript is now acceptable for publication, you may indicate that here to bypass the “Comments to the Author” section, enter your conflict of interest statement in the “Confidential to Editor” section, and submit your "Accept" recommendation.

Reviewer #1: All comments have been addressed

Reviewer #2: All comments have been addressed

2. Is the manuscript technically sound, and do the data support the conclusions?

Reviewer #1: Yes

Reviewer #2: Yes

3. Has the statistical analysis been performed appropriately and rigorously? 

Reviewer #1: Yes

Reviewer #2: Yes

4. Have the authors made all data underlying the findings in their manuscript fully available?

Reviewer #1: Yes

Reviewer #2: Yes

5. Is the manuscript presented in an intelligible fashion and written in standard English?

Reviewer #1: Yes

Reviewer #2: Yes

6. Review Comments to the Author

Reviewer #1: The authors have clearly described the research problem and the proposed solution in this revision. The previous missing details are clarified in this version.

Reviewer #2: (No Response)

7. PLOS authors have the option to publish the peer review history of their article (what does this mean?). If published, this will include your full peer review and any attached files.

Reviewer #1: No

Reviewer #2: No

---

## [Editor Report · Acceptance letter]

8 Feb 2022

PONE-D-21-18820R2 

Employing Automatic Content Recognition for Teaching Methodology Analysis in Classroom Videos 

Dear Dr. Jeon:

I'm pleased to inform you that your manuscript has been deemed suitable for publication in PLOS ONE. Congratulations! Your manuscript is now with our production department. 

Kind regards, 

on behalf of

Dr. Felix Albu 

Academic Editor

PLOS ONE